# Fast Adiabatic Mode Evolution Assisted 2 × 2 Broadband 3 dB Coupler Using Silicon-on-Insulator Fishbone-like Grating Waveguides

**DOI:** 10.3390/nano13202776

**Published:** 2023-10-17

**Authors:** Yulong Xue, Lingxuan Zhang, Yangming Ren, Yufang Lei, Xiaochen Sun

**Affiliations:** 1State Key Laboratory of Transient Optics and Photonics, Xi’an Institute of Optics and Precision Mechanics, Chinese Academy of Sciences, Xi’an 710119, China; zhanglingxuan@opt.cn (L.Z.); renyangming2017@opt.cn (Y.R.); leiyufanq2017@opt.cn (Y.L.); 2University of Chinese Academy of Sciences, Beijing 100049, China

**Keywords:** fast adiabatic, broadband coupler, grating waveguides, power splitter, local eigenmodes

## Abstract

We report a novel 2 × 2 broadband 3 dB coupler based on fast adiabatic mode evolution with a compact footprint and large bandwidth. The working principle of the coupler is based on the rapid adiabatic evolution of local eigenmodes of fishbone-like grating waveguides. Different from a traditional adiabatic coupling method realized by the slow change of the cross-section size of a strip waveguide, a fishbone waveguide allows faster adiabatic transition with proper structure and segment designs. The presented 3 dB coupler achieves a bandwidth range of 168 nm with an imbalance of no greater than ±0.1 dB only for a 9 μm coupling region which significantly improves existing adiabatic broadband couplers.

## 1. Introduction

Silicon photonics has been successfully studied and developed for various applications in both academia and industry due to its unique advantages, such as wide transparent windows, ultra-high refractive index contrast, and CMOS (complementary metal oxide semiconductor) compatibility [1,2,3,4,5,6,7,8,9]. Great efforts have been made to further improve the performance of silicon photonic devices to meet the requirements of demanding applications. One such device is a broadband 2 × 2 broadband 3 dB optical power splitter, which has been widely used as a basic element in photonic integrated circuits for its ability to split the optical power evenly into two output ports with a broadband spectrum. This device plays a critical role in many applications such as optical communication, telecommunications, sensing, and microwave photonics [10,11]. A compact low-loss 2 × 2 broadband 3 dB coupler with an ideal splitting ratio and broadband operation bandwidth is a key component for many large-scale silicon photonic systems [12,13,14,15,16]. Common methods to realize 2 × 2 power splitting include directional couplers (DCs), multimode interference (MMI), and 3 dB adiabatic couplers. A DC has a very limited bandwidth and high sensitivity to manufacturing variance. An MMI coupler also has limited bandwidth and usually non-negligible insertion loss. An adiabatic coupler (ADC), made through an adiabatic transition [17], has been proposed to provide wider bandwidth. It should be noted that an adiabatic coupler generally involves the evolution of one mode, which distinguishes it from DC and MMI couplers. The low insertion loss of an ADC can be achieved by carefully keeping the mode order unchanged throughout the adiabatic transition region. Unlike a DC, an ADC does not demand the same level of accurate control of the coupling length and can realize both low-loss and broadband operation. However, a typical ADC suffers from a very long adiabatic coupling region. Although some ideas have been proposed to shorten the length [18,19,20], simultaneous achievement of broad bandwidth and a very compact size such as 9 μm remains to be seen. A fast adiabatic coupling method was proposed earlier to reduce the length of the adiabatic coupling mode evolution interval [21]. Subwavelength grating (SWG) structures offer the flexibility to tailor the refractive index and the dispersion properties of SOI photonic devices and, thus, provide a means to reduce the device footprint. SWGs are periodic structures that have their grating periods smaller than $\frac{\pi }{\beta }$, where β is the propagation constant of the optical mode under consideration. The light propagating in an SWG-based waveguide can be treated as though it were in a regular waveguide made of a homogeneous material that has the equivalent refractive index. SWG structures can be engineered to let the light “see” less waveguide material and more cladding material. This can be used to make the optical modes less dispersive and/or less sensitive to fabrication imperfections. The fish bone is one kind of subwavelength grating structure, so the advantages of fish bone geometry compared to conventional couplers are smaller footprint, less dispersive, and less sensitive to fabrication imperfections. Combining the fast adiabatic coupling with a subwavelength grating (SWG), which is proven to provide wide bandwidth, may provide an answer. However, we found that a simple combination of ADC and SWG usually results in a large loss.

In this paper, we address the loss problem resulting from subwavelength grating (SWG) fast bending by introducing a fast adiabatic in a fishbone-like waveguide structure. The design first makes use of an effective refractive index method, in which the fishbone waveguide is approximated as a homogeneous medium, to solve the fast adiabatic transition conditions. Then, the three-dimensional (3D) finite-difference time-domain method is used for further optimization and verification. An optimized 2 × 2 fast adiabatic 3 dB coupler design with a coupling length fewer than 10 μm, a bandwidth of 168 nm, and an insertion loss of less than 0.45 dB with an imbalance of no greater than ±0.1 dB is obtained. If we relax the power-splitting ratio tolerance of ±0.4 dB, as in many published articles, the bandwidth reaches 270 nm. We believe that the design can be further improved with more sophisticated 3D optimization for even smaller size, smaller insertion loss, and larger bandwidth.

## 2. Design Principles and Simulation

A traditional adiabatic coupler generally consists of a pair of straight taper waveguides, as shown in Figure 1a, so the power transfer between modes can only be controlled through variation of the taper waveguides cross-section. Compared to DC and MMI couplers, the operating theory of adiabatic couplers (ADCs) involves only one mode, and the mode order stays the same through the adiabatic transition region even though the mode profiles at distinct locations are different. The adiabatic modes transmission can only be realized through the slow variation of the waveguide transmission cross-section, typically requiring a coupling interval of hundreds of micrometers. The fast adiabatic coupler brings tailored geometry tilt (Δy in Figure 1b) as an extra dimension to control the power transfer between modes. Both the tailored geometry tilt and the waveguide cross-section shape are designed to shorten the coupler length. Therefore, a more compact coupler is always obtained by the fast adiabatic method rather than traditional methods [22,23]. Similarly, we bring subwavelength structure as an extra dimension to further shorten the size of the coupler, as shown in Figure 1c.

Before we study the simulation steps of fast adiabatic coupling supported by using silicon-on-insulator fishbone-like grating waveguides, let us first briefly analyze the principle of traditional adiabatic coupling and confirm some basic parameters related to the traditional adiabatic coupling device we are going to use, such as W_1_, W_2_, W_3_, Λ, and L. We are improving it into a fast adiabatic coupling device supported by subwavelength based on these parameters. Here, W_1_ and W_2_ are the input waveguide widths of the adiabatic coupling device, which we can know from the earlier article [A]; the difference between W_1_ and W_2_ controls the insertion loss of the traditional adiabatic coupling device, and a bigger difference causes a smaller insertion loss, and smaller difference causes a big insertion loss. But the big difference always causes the decrease in matching between transmission waveguides and coupler waveguides, and causes energy to transfer from one mode to another, resulting in another loss. Thus, the value of the difference needs to be balanced in the above loss. Meanwhile, the width of output waveguide always defined by the transmission waveguides only needs to be kept consistent, because this can ensure a 1:1 light output efficiency. For period and duty circle, values are only selected far away from the forbidden band. Given that previous researchers have already studied this type of problem, we will not elaborate on it here. Instead, we will leverage the results of previous research [19] to expedite our own study.

There is also an important parameter coupler length that also controls the insertion loss of the traditional adiabatic coupling device; a bigger length causes smaller insertion loss, and a smaller length causes big insertion loss. This is one of the main targets we want to decrease by using fishbone-like and tailored geometry tilt waveguide, and another is insertion loss. Because the fishbone waveguide is a nonuniform media waveguide, the design method for the subwavelength structure waveguides proposed contains two steps. Firstly, the subwavelength structure waveguides are modeled and are equivalent to a normal waveguide through effective medium theory (EMT) [19]. The subwavelength structure waveguides only support zero-order diffraction [24]. The diffraction process of light is approximated as the transmission process in the homogeneous medium [25]. The refractive index was given approximately by Rytov’s formula as [26,27,28] follows:(1)$$n_{0}^{2}=p\cdot n_{1}^{2}+(1-p)\cdot n_{2}^{2}$$
where p is the duty cycle of the grating waveguide, $n_{1}$ is the refractive index of silicon, $n_{2}$ is the refractive index of silicon dioxide, and $n_{0}$ is the refractive index in equivalent waveguide belong to subwavelength structure waveguide. Applying the EMT, we discuss the conventional strip waveguide, fishbone subwavelength waveguides, and conventional SWG shown in Figure 2a–c by comparing their effective refractive indexes. Their dispersion curves of effective refractive indexes are given in Figure 2d. The key design idea of the fast adiabatic coupler device is to slice the equivalent waveguide coupling region and then obtain the best adiabatic effect by finding the specific deflection between the slices. However, the low equivalent refractive index of the fully etched subwavelength waveguide will cause large losses. There might be two reasons for this: firstly, in waveguides, light propagates through modes. Each mode has an effective refractive index associated with it. When the waveguide bends, the shape of the mode changes, leading to a decrease in the degree of match between modes. This decrease in matching will cause energy to transfer from one mode to another, resulting in loss. Secondly, the effective refractive index is a key parameter describing light propagation in waveguides. It quantifies the phase delay per unit length relative to the phase delay in a vacuum. When the effective refractive index is high, it means that the phase delay of light in the waveguide is large, so light needs more time to pass through the waveguide. This allows light more time to adapt to the bending of the waveguide, thereby reducing mode mismatch and energy loss caused by bending. The analysis of the sources of loss is very important, as it guides us to reduce loss within a shorter coupling distance.

Therefore, if a straight waveguide is added to the subwavelength waveguide, that is, using a fishbone waveguide, its equivalent refractive index becomes higher and can effectively reduce the loss caused by deflection. Although the fishbone waveguide will reduce the bandwidth of the device more than conventional SWG, it not be significantly reduced compared to the conventional SWG. The fishbone waveguide enables a balance between loss and bandwidth, so our device has low insertion loss while having high bandwidth. Therefore, the fishbone structure waveguide is chosen as the subwavelength structure in this paper. Here, the height of the silicon waveguide is 220 nm, the thickness of the SiO_2_ cladding and box layer is 2 μm, and other parameters are shown in Table 1.

As shown in Figure 3, the grating part of the fishbone waveguides coupler is equivalent to the orange medium of the traditional DC using the EMT.

The second step is to apply the fast adiabatic method to calculate and optimize the equivalent model shown in Figure 3. We know about the orthogonality condition in modes in directional coupler. This condition cannot be satisfied in waveguide structures in which the waveguide parameters vary in the direction of propagation. It cannot be satisfied in the sense that we cannot find the same base of orthogonal modes at each cross-section along x. However, we can define the local normal modes at the position $x_{0}$. The local normal mode representation will now become a function of x. According to the fast adiabatic method, the equivalent model is sliced to many connected cross-sections Cs(1)~CS(n). Then the local odd mode $\left([e_{i},h_{i}]\right)$ and even $\left([E_{i},H_{i}]\right)$ symmetric fundamental modes of all the cross-sections are calculated. The refractive index is maximum and the difference between these two modes is the smallest relative to other modes, which will cause the two modes on any two adjacent cross-sections to couple with each other, which is also the main source of adiabatic transmission loss. Therefore, we design the tailored geometry tilt of the waveguides cross-section to obtain the minimal transmission loss in the transmission direction to focus the power on only one mode (this is also the reason why it is called fast adiabatic coupling): either odd or even mode. The key transmission coefficients $\eta_{i}$ of odd (even) to even (odd) mode between two adjacent cross-sections Cs(i) and Cs(i+1) are calculated by Formula (2), which decides the size of the coupler loss we just mentioned.
(2)ηi=Re∫ei+1×Hi*⋅dS∫Ei×hi+1*⋅dS∫ei+1×hi+1*⋅dS1Re∫Ei×Hi*⋅dS

The tailored geometry tilt between Cs(i+1) and Cs(i) Δy((i,i+1)) can be optimized to obtain the minimum $\eta_{i}$. Finally, the $\eta_{1}-\eta_{n}$ can be calculated, and the fishbone structure is designed when a step is given. The method of Formula (2) can be found in the book by A.W. Snyder and J. Love [29].

Using the method introduced above, the broadband 3 dB coupler is designed as shown in Figure 4. The coupler consists of three regions. Region I is the taper that linearly converts the upper input waveguide width W_1_ to the upper optical waveguide width W_core_ and the lower waveguide width W_2_ to the lower optical waveguide width W_core_ of the coupling part. The input taper ensures that the light is smoothly converted to the first supermode and is adiabatically transferred to the coupling region. Region II is the mode adiabatic evolution region, and the overall width of the two fishbone waveguides are converted from $\left(W_{1},W_{2}\right)$ to $\left(W_{3},W_{3}\right)$. Region III is the output taper, which allows the supermodes output from the coupling region to be adiabatically distributed to the output waveguide. Meanwhile, all the gaps of the three regions are the same. The basic parameters are given in Table 1.

The key parameters of this coupler are the tailored geometry tilts $\Delta y_{1}-\Delta y_{n}$ of region II. The scanning step is 2 nm, and then $\Delta y_{1}-\Delta y_{n}$ is calculated and shown in Figure 5, as follows. $\Delta y_{1}-\Delta y_{n}$ is the tailored geometry tilts, $\sum_{x\cdot n}\Delta y(i)$ is the sum of the tailored geometry tilts, and then x is the ratio of the distance traveled by light after entering the coupling region to the total length of the coupling region. This also is the reason why our coupler region is curved, not straight. In our simulations, we used the refractive index data reported by Palik [30] for both Si and SiO_2_. We used Meep [31] as the main tool for this simulation. The three-dimensional finite-difference time-domain (3D FDTD) method with perfectly matched layer conditions applied in the mode propagation direction [32,33,34] is used to verify it. In addition, our computer’s CPU has 6 cores, 12 logical processors, and a base speed of 2.59 GHz. Running a multiwavelength simulation takes about 6 h.

## 3. Results and Discussion

The top view of the simulated electric field distributions of the even and odd TE supermodes for a fishbone adiabatic 3 dB coupler is shown in (Figure 6a) and (Figure 6b) when light is injected at the lower port and upper port when the light wavelength is 1.55 μm.

The broadband property of the coupler is calculated in the same way. Due to the asymmetry of the coupler design, the broadband power-splitting ratios of two injecting modes are different. As shown in Figure 7a, when light is injected from the lower port, the power-splitting ratio of −3 dB ranges from 1510 nm to 1678 nm with an imbalance of less than 0.1 dB, and the corresponding Figure 7b shows another injecting mode, where the power-splitting ratio of −3 dB ranges from 1500 nm to 1678 nm with the same imbalance. Insertion losses of the two injecting modes are given in Figure 7c, and the coupler maintains an insertion loss less than 0.45 dB. The coupling length is as short as only 10 μm.

The fabrication tolerances are also explored through simulating our device for process limit, by considering variations in waveguide thickness (ΔH) and other feature sizes containing ΔW and Δl. ΔW is the variation applied to both the total waveguide width and the strip width of the two fishbone waveguides transitions from $\left(W_{1},W_{2}\right)$ to $\left(W_{3},W_{3}\right)$ in the mode adiabatic evolution region (region II in Figure 4). Δl is the variation applied to the fishbone fin length. Gaps between the two waveguides and the grating period were maintained as constants without fabrication errors. To account for possible fabrication process variations that may occur in commercially available SOI fabrication processes [35], we consider ±10 nmas fabrication errors, so ΔH, ΔW, and Δl were set in the range of [−10, 10] nm in our simulations and discussion. The splitting ratios were calculated with 0 nm, 10 nm, and −10 nm variations of the structure parameters. The results are shown in Figure 8, where it can be seen that the splitting ratios imbalance expand from 0.1 dB to 0.4 dB. Meanwhile, the insertion losses were calculated under the same situation and are shown in Figure 9. The insertion loss of device changes from 0.45 dB to 1.2 dB. Table 2 compares our best design-simulated bandwidth of operation, when better than 3 ± 0.1 dB, to other reported couplers in the literature, which shows the advantage of our device.

.

## 4. The Analysis about Device with Defects

Finally, the presence of defects within the grating structure was considered. We studied the single and double defects in our device shown in Figure 4, and the results are listed in Table 3. The “Single” means that the device has one consecutive fin, and the “Double” means that two consecutive fins are missing, far away from the gap of two waveguides. Then, we can see the minimal impact of the device when the size of the defect is less than 50 nm in the “Single” part and 30 nm in the “Double” part of Table 3. 

Due to the low configuration of our computers, the simulation time for a single device is quite long. Therefore, we did not conduct large-scale random testing. Instead, we verified the performance of the device under eight groups of defects with multiple random positions and sizes. The results consistently indicate that the maximum size of the gate missing determines the size of the loss; whether the position of the gate missing is between two grating waveguides and whether there are multiple missing gates connected determines the size of the bandwidth. However, the good news is that we slightly lowered our requirements and obtained a bandwidth of no less than 40 nm according to the 3 ± 0.4 dB splitting ratio range. Most devices show a bandwidth of hundreds of nanometers, which is sufficient to prove the high tolerance of our devices under the above verification. Multiple defects are randomly located and sized within the coupling interval, the maximum defect size is 90 nm, and the maximum number of connected defects is four. The location of missing fins is any location of coupling interval.

We show one kind of the random distributions of a defect size and location in Figure 10. The sizes of these defects with orders in Figure 10 are [61, 27, (40, 13), 50, 43] nm. Two sizes of defect of three in the matrix affects two fishbone-like waveguides but we still consider this a defect.

## 5. Conclusions

In conclusion, we introduce a novel 2 × 2 broadband 3 dB coupler design by combining the fast adiabatic method with fishbone-like subwavelength waveguide and achieve a very compact and ultra-broadband performance. The adiabatic coupling length is a record low of 9 μm operating at such a wide bandwidth of 168 nm with an imbalance of no greater than ±0.1 dB. The maximum insertion loss of the even supermodes is less than 0.07 dB. At the same time, our device has a certain robustness to the existence of defects. We believe that the performance can be further optimized and the design concept can be extended to other photonic devices.

## Figures and Tables

**Figure 1 nanomaterials-13-02776-f001:**
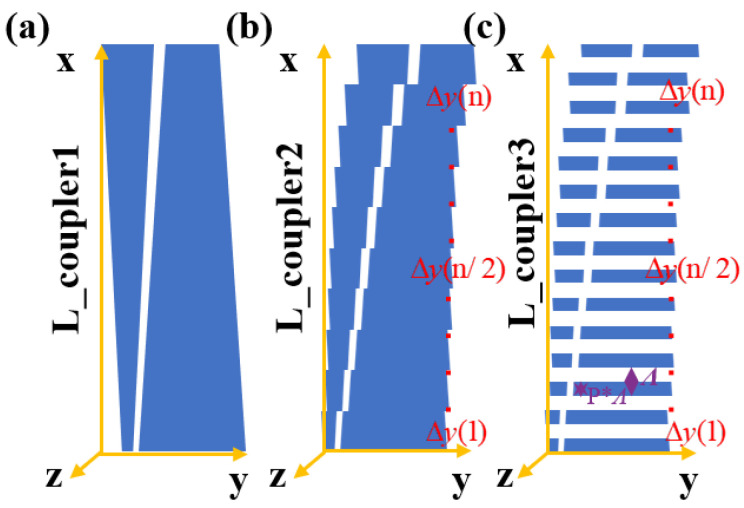
Schematic diagrams for (**a**) traditional adiabatic coupler, (**b**) the fast adiabatic coupler, and (**c**) the fast adiabatic coupler assisted fishbone structure. L_coupler3 < L_coupler2 < L_coupler1.

**Figure 2 nanomaterials-13-02776-f002:**
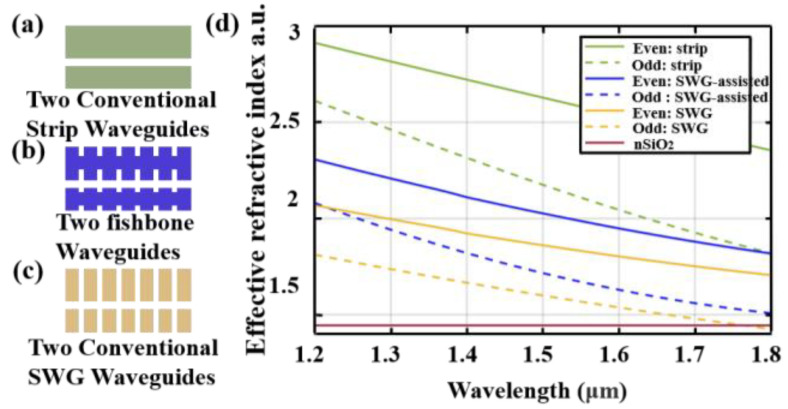
Schematic diagrams for (**a**) two conventional strip waveguides, (**b**) two SWG-assisted strip waveguides (fishbone waveguide), and (**c**) two conventional SWG waveguides. (**d**) Calculated effective indices of the lowest-order even and lowest-order odd TE supermodes as functions of wavelength for the waveguides depicted in (**a**–**c**).

**Figure 3 nanomaterials-13-02776-f003:**
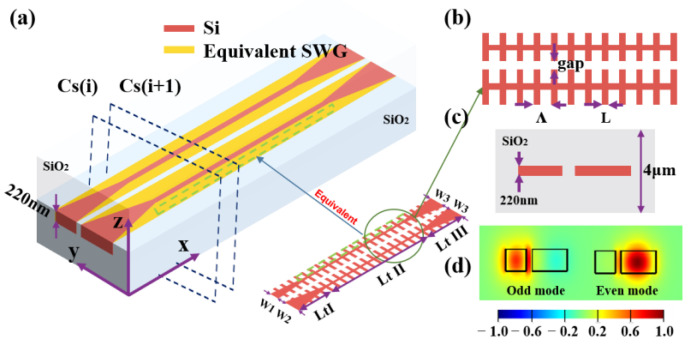
(**a**) Equivalent model of fishbone waveguide. (**b**,**c**) Important parameters of the fishbone waveguide. (**d**) Odd mode and even mode of injection cross-section.

**Figure 4 nanomaterials-13-02776-f004:**
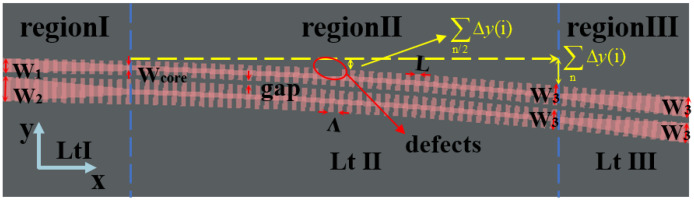
Fast adiabatic mode evolution-assisted broadband 2 × 2 broadband 3 dB coupler using silicon-on-insulator fishbone-like grating waveguides (lengths of region I and region III are compressed).

**Figure 5 nanomaterials-13-02776-f005:**
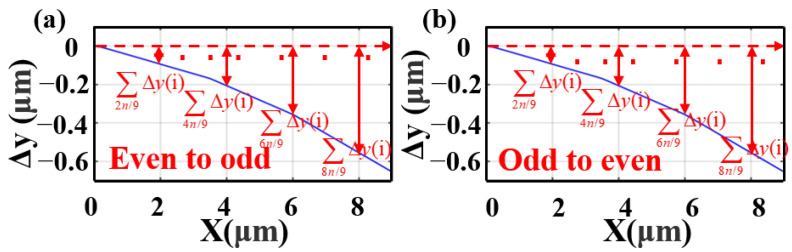
(**a**) Δy of mode 1 and mode 2; (**b**) Δy of mode 2 and mode 1; 1, 2 is the order of the determined mode profiles on Cs(i) and Cs(i+1).

**Figure 6 nanomaterials-13-02776-f006:**
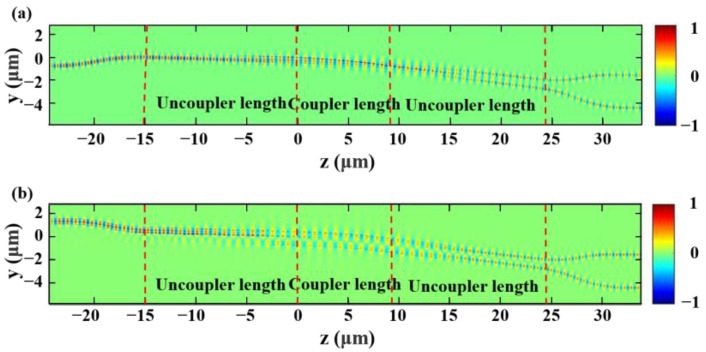
The simulated transmission electric field distributions of (**a**) the upper port (even TE supermode); (**b**) the lower port (odd TE supermode) when the input wavelength of the 2 × 2 coupled devices is 1.55 μm.

**Figure 7 nanomaterials-13-02776-f007:**
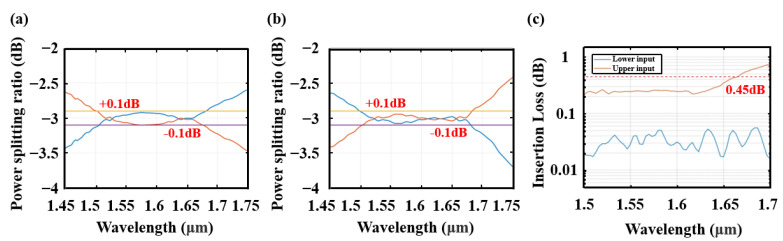
(**a**) The splitting of 2 × 2 coupled devices when the odd TE supermode was injected. (**b**) The splitting of 2 × 2 coupled devices when the even TE supermode was injected. (**c**) Insert loss of the device when two kinds of modes were injected.

**Figure 8 nanomaterials-13-02776-f008:**
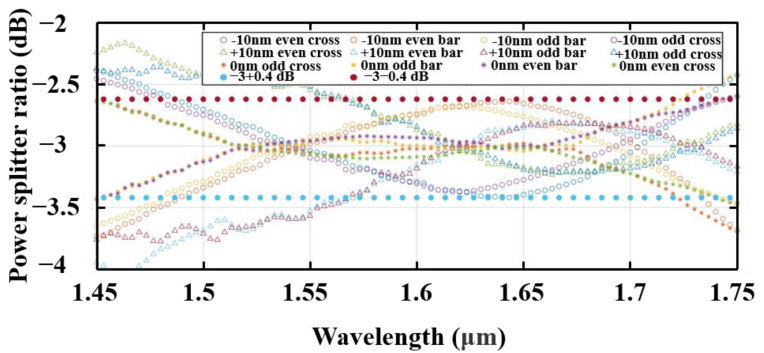
Simulated power-splitting ratio of the SWG-assisted fast adiabatic coupler with some fabrication errors. $\Delta h=\Delta W_{1}=\Delta W_{2}=\Delta W_{3}=\Delta W_{core}=\Delta l\pm 10\;nm$.

**Figure 9 nanomaterials-13-02776-f009:**
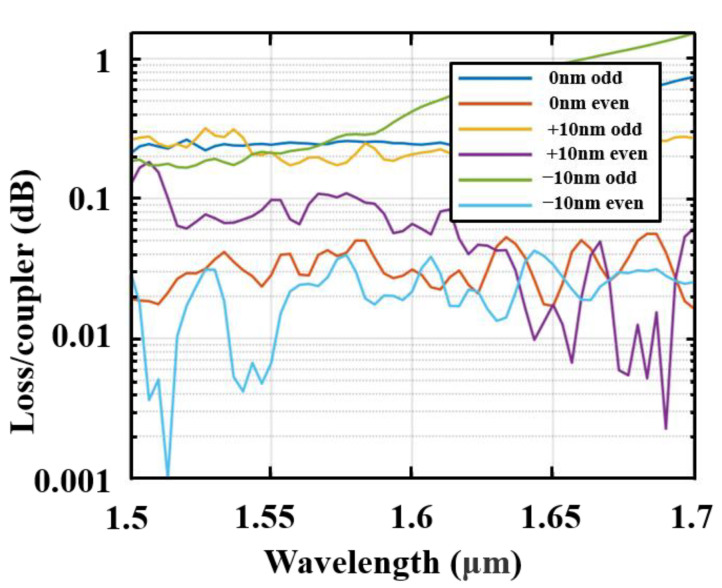
Simulated insertion loss of the SWG-assisted fast adiabatic coupler with some fabrication errors. $\Delta h=\Delta W_{1}=\Delta W_{2}=\Delta W_{3}=\Delta W_{core}=\Delta l\pm 10\;nm$.

**Figure 10 nanomaterials-13-02776-f010:**
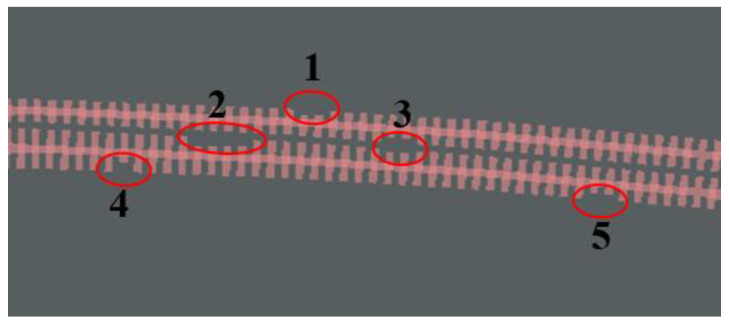
The random distribution of a defect size and location in a coupler region.

**Table 1 nanomaterials-13-02776-t001:** Parameters of fishbone structure waveguide.

Parameters	W_1_	W_2_	W_3_	W_core_	LtI	LtII	LtIII	Gap	Λ	L
unit/μm	0.35	0.61	0.48	0.12	15	9	15	0.1	0.2	0.1

**Table 2 nanomaterials-13-02776-t002:** Comparison of our device with other simulation-based devices’ performance in coupler length (μm) and bandwidth (nm) in the interpretation.

The splitting ratio is unified to 3 ± 0.1 dB	7	10	[36]	Subwavelength coupler
	27	8	[37]	Directional coupler
	3250	15	[38]	Directional coupler
	16	80	[39]	Adiabatic coupler
	13	20	[13]	Bent directional couplers
	7	42	[40]	Curved directional couplers
	7	46	[41]	Curved directional coupler
	34	21	[42]	Directional coupler
	19	20	[33]	Subwavelength coupler
	23	30	[14]	Multimode interference coupler
	9	168	Our work	Fishbone-like fast adiabatic coupler

**Table 3 nanomaterials-13-02776-t003:** The performance of our device in bandwidth (nm) and loss (dB) in the interpretation when the grating misses the fins with different size (nm) and single/double number.

The splitting ratio is unified to 3 ± 0.1 dB	10	167/167	0.052/0.561	170/176	0.052/0.592
	20	160/166	0.053/0.591	153/160	0.053/0.596
	30	156/163	0.054/0.587	133/130	0.054/0.600
	40	140/140	0.055/0.561	103/53	0.056/0.608
	50	127/123	0.057/0.562	67/27	0.056/0.623
	60	113/113	0.058/0.564	~	
	70	107/60	0.060/0.565	~	~
	80	100/57	0.061/0.569	~	~

## Data Availability

Data underlying the results presented in this paper are not publicly available at this time but may be obtained from the authors upon reasonable request.

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
