# Peer review of "Fast Adiabatic Mode Evolution Assisted 2 × 2 Broadband 3 dB Coupler Using Silicon-on-Insulator Fishbone-like Grating Waveguides"

_nanomaterials, 2023, doi:10.3390/nano13202776_

Round 1

Reviewer 1 Report

In this paper, a broadband low-loss 2 x 2 coupler is proposed by applying a fast adiabatic method to a fishbone-shaped subwavelength grating waveguide. Each of the principles considered is supported by previous work by various groups. The conclusions are based on analytical calculations and numerical simulations and have high scientific validity. The novelty and impact of the proposed device is recognized by its small size, expected broadband performance, and high error tolerance compared to the devices in previous studies. Therefore, I judged that this paper should be published in this journal after answering my minor questions below and revising the manuscript.

1) The MZI is described over lines 32-44 of the paper. This is intended to demonstrate the usefulness of the 2 x 2 coupler, but is nevertheless very redundant. I think it is better to omit or delete this explanation since MZI itself is not addressed in this paper.

2) In lines 47-48, the abbreviations "directional couplers (DC)" and "multimode interference (MMI)" appear. The abbreviation "DC" should be used for "directional couplers" in line 48, and "MMI" should be used for "multimode interference" in line 49. 

Should the "adiabatic 3 dB coupler" in line 48 be "3-dB adiabatic coupler (ADC)" ?

3) As an interpretation of Fig. 2, the line 102-104 say "The fishbone and conventional SWG waveguide express flatter dispersion than conventional strip waveguide which has the potential to bring better broadband device performance." 

The vertical axis is absolute, however, in the case of the effective refractive index, isn't the rate of change is important, not the amount of change? If we look at the rate of change, isn't the fishbone the same or inferior to the others?

4) Lines 106-107 say "the low equivalent refractive index... will cause large losses". Why is this the case?

5) The sentence in lines 172-174 is grammatically incomplete.

6) In lines 181-182 and 183-184, "the power splitting ratio range from..." does not make sense. Does it mean "the power splitting ratio of -3 dB ranges from..."?

Author Response

Thanks for your serious and patient review. We appreciate your valuable comments very much. All your comments are considered seriously and our manuscript has been revised carefully in accordance with your comments where possible. Please find the detailed responses in the attachment and the corresponding revisions highlighted track changes in the re-submitted files. The revised parts are marked with RED font in the revised version. 

Reviewer 2 Report

This is a single blind review of manuscript nanomaterials-2641785 titled "Fast adiabatic mode evolution assisted 2 × 2 broadband 3 dB coupler using silicon-on-insulator fishbone-like grating wave-guides" by Yulong Xue et al. The paper is submitted to the special issue "Nanostructured Materials for Photonic and Plasmonic Applications".

The work presented in the manuscript is the numerical design and simulated evaluation of an integrated waveguide 2x2 power splitter (3dB coupler), based on SOI technology. The core idea is to replace straight adiabatic waveguides with fishbone ones (the waveguide width is periodically modulated in the transversal direction), as this improves bandwidth and minimizes the footprint at little expense to insertion loss. The design firstly makes use of the effective-index/medium theory and then numerical 3D-FDTD simulation. The design is compared to similar designs from the literature, in terms of power splitting imbalance and insertion loss. A parameter sensitivity analysis is also presented, in order to anticipate fabrication tolerances.

The paper is inside the scope of the SI of Nanomaterials and could be considered for publication after improvements have been made.

A. All the designs and comparisons in Table II are against classical approaches to directional couplers. Have the Authors considered newer approaches, e.g., the inverse designed ones, e.g., by J. Vuckovic group in Stanford? Such designs provide ultra-compact footprints and optimal configuration at selected wavelengths. Also, Table II should include more details, e.g., the technology platform used, the distinct feature of each design, or if the metric is simulated or measured.

B. It is not clear how the Authors came about the design parameters, e.g., in Table I. Are these an outcome of some optimization process, or are they picked from literature and fine-tuned? If an optimization was done, which were the steps and the criteria/targets? Also, it is not clear why the coupler region (II in Fig. 4) is arced; is that a necessity or a choice?

C. These grating waveguides are very sensitive to defects, as is exhibited by the sensitivity analysis results in Table III. For that matter, even though several cases are studied, all of them are "deterministic" and localized; in order to better anticipate the true fabrication tolerance, I recommend doing a stochastic analysis, e.g., introducing multiple defects and/or various parameter variations in a *random* manner.

D. Other improvements & comments:

D01. Fig. 1 needs improvements in its annotation. Also the xyz axes should be marked.

D02. In Eq. (1), describe what each symbol/parameter is.

D03. In line 116 (referring to Table I), I believe that a schematic is needed to mark these dimensions, both in top-view and in cross-section view.

D04. In line 123 a "grey medium" is mentioned. What is this? The yellow region in Fig. 3?

D05. In lines 126-141, a visualization of the even/odd supermodes of coupler in a figure (even an abstract figure with 1D curves) would help the readers better understand the design strategy.

D06. Please provide a reference for Eq. (2).

D07. Fig. 4 needs improvement as some annotation is not visible. For example the "II" (in Region II) is different font than "I" and "III".

D08. Was a commercial software used for 3D-FDTD? Please provide reference, especially if an in-house or open-source (community) one was used. Also, the simulation time and machines used should be indicated.

D09. In Fig. 5, what is the 1/9 factor in the sum? Is this a physical quantity (e.g. the 9 microns in the length) or some discretization of the SWG (in 9 periods)? Please clarify what this is, and if it a design choice or an outcome of optimization.

D10. In line 205 "measured" is mentioned. Please correct to "simulated".

D11. Table III is a little confusing. Suggested improvements: It should be clarified that "single/double" refers to missing fins in the grating waveguide (in the double case, it should be clarified if two consecutive fins are missing, or opposite side etc.); the text in column 1 (interpretation) should be moved to table caption; the last column should be made first. 

D12. In Table III, there are some weird results. For the single case, the Authors get a non-monotonic change in IL (dB) when the defect size increases 10-->50 nm. For the double case, the IL (dB) is marginally affected, i.e., every value is below 0.1 dB. Please provide convincing explanations/discussions about these findings (or correct).

E. Please use math environment for variables/symbols in the in-line text (variable symbols in display equations and in figures should be in the same font as the ones shown in the text).

Language is OK for understanding the core aspects, but certainly needs improvement, especially in the subtle aspects of the work.

Author Response

(The authors gave the same response as above.)

Reviewer 3 Report

In this short paper authors present a 2 × 2 broadband 3dB coupler designed by combining the fast adiabatic method with a fishbone-like subwavelength waveguide. This is a simulation work. As such it is sound and meaningful. However in the practical implementation problems may arise, other than the ones well considered in the simulation, that may reduced the performance of the coupler. Please add a section discussing these issues at least in a introductory way. Furhter work is advised before drawing conclusions about the usefulness of the approach.

Some obvious writing mistakes can be easily corrected. A general polishing of the text may help readability

Author Response

(The authors gave the same response as above.)

Reviewer 4 Report

This article reports on fishbone-like. silicon-on-insulator It would be worthy of publication if the following points were corrected.

Regarding Figure 1, the fish bone model used in this study, what are the advantages of this geometry compared to conventional couplers?

The fishbone-shaped cavities are aligned diagonally on the longitudinal axis, what is the reason for not making them straight?

For Equations 1 and 2, please add what each term means.

The text in Figures 2, 6 and 8 is small, please make it larger.

Table 2, which one is (a): the traditional adiabatic coupler (b): the fast adiabatic coupler (c) the 90 fast adiabatic coupler? Please write in the table to help the reader understand.

It is well written. It is worthy of acceptance if the sections I commented on to the author can be corrected.

Author Response

(The authors gave the same response as above.)

Round 2

Reviewer 2 Report

The Authors have addressed most of the minor comments of my first review, but only "circumvented" (failed to convincingly tackle or address) the important ones. So, after its revision, this paper remains an interesting simulation-only design, with average novelty and potential impact.

Some minor comments on the revised version:

A. The units should be in upright (not italic) font.

B. The Tables II and III are not optimized for space and readability. As I said in my previous review, the text in the first column is best fitted to be in caption of the table. Also, the column headers are weirdly placed. Finally, avoid decimals when not needed (all columns should have the same number of decimal digits).

Improvements have been made. The response letter, however, was also hard to comprehend at times...

Author Response

Thanks for your serious and patient review. We appreciate your valuable comments very much. All your comments are considered seriously and our manuscript has been revised carefully in accordance with your comments where possible. Please find the detailed responses in the attachment and the corresponding revisions highlighted track changes in the resubmitted files. The revised parts are marked with RED font in the revised version. 
